# Endoscopic Retrieval of Esophageal and Gastric Foreign Bodies in Cats and Dogs: A Retrospective Study of 92 Cases

**DOI:** 10.3390/vetsci10090560

**Published:** 2023-09-05

**Authors:** Giulia Maggi, Mattia Tessadori, Maria Luisa Marenzoni, Francesco Porciello, Domenico Caivano, Maria Chiara Marchesi

**Affiliations:** 1Department of Veterinary Medicine, University of Perugia, Via San Costanzo 4, 06126 Perugia, Italy; giulia.maggi@studenti.unipg.it (G.M.); marialuisa.marenzoni@unipg.it (M.L.M.); francesco.porciello@unipg.it (F.P.); 2“Ponte Felcino” Veterinary Clinic, Via della Ghisa 3, 06134 Perugia, Italy; tessadorimattia@gmail.com

**Keywords:** endoscopy, foreign bodies, dogs, cats, gastric, esophagus

## Abstract

**Simple Summary:**

Esophageal and gastric foreign bodies commonly occur in small animal practices. Medical records of cats and dogs undergoing endoscopic removal of esophageal and gastric foreign bodies have been reviewed to evaluate the factors that can influence the success rate and timing of the procedure. Ninety-two animals were included in the study. Endoscopic removal of foreign bodies was successful in 88% of cases, and the mean time spent for the extraction was 59.74 min (range, 10–120 min). The success rate and timing of endoscopic foreign body removal can be influenced by several factors including the size and age of the animals, the localization of foreign bodies, the device used, and the operator’s experience.

**Abstract:**

Esophageal and gastric foreign bodies (FBs) commonly occur in small animal practices, and their endoscopic removal has been previously reported. However, few studies reported the endoscopic instruments used for the retrieval attempt and the time spent for endoscopic removal. Therefore, the aim of this study is to evaluate the factors that can influence the success rate and timing of the endoscopic retrieval of FBs. The medical records of 92 animals undergoing endoscopic removal of esophageal (n = 12) and gastric (n = 84) FBs have been reviewed. Two dogs had FBs in both the esophagus and stomach. From medical records and video recordings, there were extrapolated data on signalment, clinical signs, endoscopic devices used, success of retrieval, and duration of endoscopy. Endoscopic removal of FBs was successful in 88% cases, and the mean time spent for the extraction was 59.74 min (range, 10–120 min). The success rate and timing for the removal of endoscopic foreign bodies (EFBs) are influenced by several factors in our population: medium-breed dogs, adult animals, and localization of FBs in the body of the stomach increased the probability of failure during the endoscopic retrieval attempt. Conversely, the success and timing of the retrieval of EFBs were higher in puppies and with increasing operator’s experience. Moreover, the use of combination devices such as polypectomy snare and grasping forceps negatively influenced the success of extraction of FBs. Further prospective and comparative studies in a large and multicentric population of patients can be useful to create interventional endoscopic guidelines, as in human medicine.

## 1. Introduction

The ingestion of foreign bodies (FBs) is relatively common in both canine and feline patients, although dogs are more likely to present with esophageal and gastric FBs than cats [1]. Esophageal FBs are mostly reported in small dogs because of the size of their esophagus, and the obstruction is usually localized where natural narrowing of the organ occurs (thoracic inlet, heart base, and caudal esophageal region) [2]. Bone or cartilage material, fishhooks, balls, toys, needles, hair bands, and hairballs are frequently reported as esophageal and gastrointestinal FBs in small animals [3]. The severity of the clinical signs depends on several factors such as localization (esophageal, gastric, or duodenal), size of FBs and size of the animal, type of FBs (if traumatic or not for gastrointestinal mucosa), and duration of obstruction [1,2]. Common clinical signs are salivation, retching, gagging, vomiting, regurgitation, anorexia, pain, and respiratory distress [2,3].

The percentage of FBs that pass through the gastrointestinal tract without requiring any treatment is unknown [3]. If unresolved, FBs can become life-threatening and may need endoscopic or surgical retrieval. The prevalence of FBs requiring endoscopic removal was reported to be 0.47–0.67% in a canine hospital population [4]. The success rate for the endoscopic removal of esophageal FBs was 68–88%, whereas gastric FBs showed a success rate of 78–94% [4]. If endoscopic retrieval fails, the FB needs surgical removal [4,5]. Esophageal FBs can be pushed into the stomach. This is useful because food material (bone, cartilaginous) can be digested, whereas other FBs can be removed via gastrotomy, which has a lower rate of complication and better prognosis than esophageal surgery [2,5].

The removal of endoscopic foreign bodies (EFBs) is a minimally invasive technique showing a high success rate. Several factors influence the choice of the approach for the retrieval of EFBs, such as the type of FBs, location, and endoscopic equipment. A lot of endoscopic instruments are available in the working channel of an endoscope, including a variety of extraction forceps such as grasping forceps, alligator forceps, rat tooth forceps, baskets, and polypectomy snares [6]. Moreover, laparoscopic rigid forceps can be used coaxially to the working channel of the endoscope for the removal of esophageal FBs [6]. The time taken to attempt the esophageal or gastrointestinal retrieval of FBs can be affected by the operator’s experience, complications, or clinical condition of the animal. Previous studies have investigated success rate, risk factors, and complications [4]; however, to the best of our knowledge, few studies have reported the endoscopic instruments used and the time spent for endoscopic removal [4]. Moreover, endoscopic technique is characterized by a learning curve, and the operative skills of endoscopists increase progressively with the number of procedures.

Therefore, the aim of this retrospective study is to report data on the signaling risk factors, endoscopic techniques and instruments, success rates, and timing for the retrieval of EFBs in dogs and cats with esophageal and/or gastric FBs. We have also evaluated the correlation between the operator’s experience and the success rate for the removal of EFBs. All this information can be useful for increasing the success rate and decreasing the time for the removal of EFBs.

## 2. Materials and Methods

### 2.1. Data Collection

Medical records of dogs and cats with suspected esophageal or gastric FBs referred to the Veterinary Teaching Hospital of Perugia University and “Ponte Felcino” Veterinary Clinic for removal of EFBs between 2017 and 2022 were reviewed. Ninety-two dogs and cats with esophageal (*n* = 12) and gastric (*n* = 82) FBs were included in the study. Two dogs had FBs in both the esophagus and stomach. Data reviewed from electronic medical records included signalment (animal species, breed, age, sex, and weight), clinical signs at presentation, successful removal, endoscopic examination performed by operator, and complications. Moreover, data on type, number, and location of FBs; endoscopic instruments; time spent on the extraction or spent attempting FB retrieval before recommending surgery; and mucosa status after removal of EFBs were collected by reviewing medical records and video recordings. Ancillary devices could be used in the working channel of the endoscope (grasping forceps, polypectomy snare, rat tooth forceps, alligator forceps, and biopsy forceps) or coaxially to the endoscope (laparotomic forceps). The extraction time was considered as the time from introduction of the endoscope into oral cavity to its extraction.

All data collected from electronic medical records and video recordings were divided into subcategories to facilitate statistical analysis. Based on weight and standard of breed, animals were classified as small breed (<10 Kg), medium breed (10–25 Kg), or large breed (>25 Kg). Based on age, animals were categorized into 4 classes: puppies (age under 6 months), young animals (age between 6 and 18 months), adults (age between 18 and 120 months), or older (age over 120 months). Based on their potential for damaging the gastrointestinal mucosa, FBs were classified as non-penetrating when characterized by a smooth outer surface and as penetrating when characterized by an irregular or sharp surface. The location of FBs was categorized as esophageal or gastric. Esophageal location was further subcategorized into the cervical (from oropharynx to the thoracic inlet) or thoracic (between the thoracic inlet and the lower esophageal sphincter) region. Gastric location was further subcategorized into the fundus, body, or antrum region (Figure 1) [6].

Clinical signs were categorized as gastrointestinal, respiratory, or unspecific. Based on experience, operators were classified as novice (<50 endoscopic examinations for a year), middle (between 50 and 150 endoscopic examinations for a year), or expert (>150 endoscopic examinations for a year). Time spent for the extraction was recorded and was categorized as excellent (less than 30 min), good (less than 60 min), acceptable (less than 90 min), or unacceptable (more than 90 min). Complications were categorized as minor or major. Minor complications were characterized by inflammation of esophageal or gastrointestinal mucosa with evident hyperemia, edema, and minor bleeding. Major complications were characterized by mucosal erosion/ulceration, application of Percutaneous Endoscopic Gastrostomy (PEG), wall perforation, and death.

### 2.2. Statistical Analysis

A descriptive analysis was performed to characterize the study population. The variables were analyzed in relation to the success of removal and the time spent for extraction (excellent and excellent/good). The differences in continuous and categorical variables were compared using the paired *t*-test and χ^2^ test, as appropriate. In the univariate model, variable scoring *p* ≤ 0.05 was statistically relevant.

Variable scoring *p* ≤ 0.20 in the univariate model or considered to be biologically relevant were included in the multivariable model. Multiple models were considered in relation to the success of removal and the time spent for extraction (excellent and excellent/good). Odds ratios (OR) and corresponding 95% confidence intervals (95% CI) were obtained through means of logistic regression [7]. Data were analyzed via commercial software R, version 2.8.1 (R, Development Core Team, 2007). A value of *p* ≤ 0.05 was considered significant for the analysis.

## 3. Results

### 3.1. Animals

A total of 86 dogs and 6 cats were referred for the ingestion of FBs. The breeds of the dogs included in the study are reported in Table 1. Feline breeds included in the study were Bengal (*n* = 1) and Domestic Shorthair (*n* = 5).

A total of 28 dogs were females (32.6%) and 57 were males (66.3%). Dogs neutered were 11/86 (30.2%), 6 (21.4%) females and 5 (8.8%) males. All cats were females, and 33.3% of them were neutered (2/6). For dogs, the mean age was 50.68 ± 24 months (range, 3–204 months): 9 dogs (10.5%) were puppies, 24 (27.9%) were young, 37 (43%) were adults, and 16 (18.6%) were older. For cats, the mean age was 21 ± 18 months (range, 6–48 months): three cats (50%) were young, and three (50%) were adults. Based on body weight, 33 dogs (38.4%) were large breeds, 22 (25.6%) were medium breeds, and 31 (36%) were small breeds. All cats were classified as small breeds.

### 3.2. Clinical Signs

Clinical signs were reported in 42.4% of cases (39/92), and in the remaining 57.6% (53/92), no clinical signs were reported (ingestion of FBs was seen by owners). Clinical signs were described in 37/86 dogs (43%) and in 2/6 cats (33.3%). Clinical signs were gastrointestinal signs in 35.9% of cases (33/92), respiratory signs in 2.2% (2/92), and were not specific in 12% (11/92). Animals with clinical signs had a mean of three clinical signs (range 1–5). All clinical signs are reported in Table 2. In cats, only gastrointestinal signs (vomiting and abdominal pain) were reported. In all cases, radiographic and/or ultrasonographic examinations were performed by referring veterinarians or in the hospital/clinic of the authors, and their results suggested the presence of esophageal or gastrointestinal FBs.

### 3.3. FBs (Type, Number, and Localization)

#### 3.3.1. Dogs

The EFBs were socks (11/86 (12.8%)), plastic fragments (11/86 (12.8%)), rags (10/86 (11.6%)), bones (7/86 (8.1%)), pinecones (6/86 (7%)), wooden toothpicks (5/86 (5.8%)), needles (2/86 (2.2%)), fishhooks (2/86 (2.2%)), and other FBs (range 1–4 (1.1–4.3%)). These were penetrating FBs in 40 patients (46.5%), non-penetrating FBs in 43 patients (50%), and both in 3 patients (3.5%). The mean number of FBs in the same dog was 1.85 ± 2.28 (range, 1 to 18). In dogs, 11.7% of the FBs were localized in the esophagus, 83.8% in the stomach, and 2.4% in both. In two dogs (2.3%), FBs were localized in the stomach extending into the duodenum. The localization of esophageal and gastric FBs in dogs is summarized in Table 3.

#### 3.3.2. Cats

Four types of gastrointestinal FBs were found in the cats included in the study, and they were needles (2/6 (33.3%)), threads (2/6 (33.3%)), shoelaces (1/6 (16.7%)), and stuffed toys (1/6 (16.7%)). Penetrating FBs were reported in two patients (33.3%) and non-penetrating in four patients (66.7%). No multiple FBs were observed in the feline patients. The localization of gastrointestinal FBs in cats was only gastric—in the body of the stomach in five patients (83.3%) and in the antrum in one case (16.7%). Clinical and endoscopic findings in cats with FBs are resumed in Table 4.

### 3.4. Endoscopic Removal (Operator’s Experience, Endoscopic Instruments, Timing for the Extraction, and Success Rate)

Endoscopy was used to successfully extract FBs in 88% (81/92) of animals (75/86 dogs (87.2%) and 6/6 cats (100%)). In 11/92 (12%) patients, surgical intervention was required to safely remove FBs. The operator’s experience was categorized as novice in 16 cases (17.3%), middle in 41 cases (44.5%), and expert in the remaining 35 cases (38.1%). For the removal, various ancillary devices were used, from a minimum of one to a maximum of three. In cats, multiple endoscopic instruments for removal were not used. In 56/92 cases (60.9%) only one ancillary device was used, and multiple instruments for the extraction were used in 36/92 cases (39.1%). Ancillary devices (polypectomy snare, rat tooth forceps, alligator forceps, grasping forceps, and biopsy forceps) were used within the working channel of the endoscope, whereas laparotomic forceps were used coaxially to the operating whip. Endoscopic instruments used during the retrieval of EFBs are summarized in Table 5 and Figure 2.

Mean time spent for removal was 54.79 min ± 2.44 with a minimum of 10 min and a maximum of 120 min (Figure 3). Time spent for the extraction was excellent in 22/92 cases (23.9%), good in 35/92 cases (38%), acceptable in 28/92 cases (30.4%), and unacceptable in 7 cases (7.6%).

### 3.5. Complications

Complications were reported in 40/92 cases (43.4%), whereas no complications were observed in 52/92 (56.6%) patients. Minor complications were observed in 37/40 (92.5%) animals, whereas major complications were observed in 3/40 (7.5%) patients. Esophageal or gastric mucosal inflammation (minor complication) was present in all cases (37/37 (100%)). Major complications were esophageal mucosal erosion/ulceration in 2/3 cases (66.6%) (Figure 4) and esophageal perforation with pneumothorax and death in 1/3 patients (33.4%). Major complications were reported only for penetrating esophageal FBs (bone) in small dogs (FBs localized in the thoracic esophagus). Clinical and endoscopic findings in dogs with esophageal FBs are reported in Table 6.

### 3.6. Univariate Analysis

All data collected were analyzed with univariate analysis. The variable positively and significatively correlated to the success rate of the removal of EFBs (*p* ≤ 0.05 and OR > 1) was good time spent for extraction. Variables negatively correlated to the success rate of the removal of EFBs (*p* ≤ 0.05 and OR < 1) were medium breed, use of polypectomy snare, use of grasping forceps, use of multiple forceps, excellent time spent for extraction, and not acceptable time spent for extraction. The variable with *p* ≤ 0.2 and OR > 1 was expert operator. Variables with *p* ≤ 0.2 and OR < 1 were large breed adult, localization in thoracic esophagus, middle operator, use of laparotomic forceps, and acceptable time spent for extraction. Univariate analysis results related to success rate of the removal of EFBs are summarized in Table 7.

Variables positively and significatively correlated to excellent time spent for extraction of EFBs (*p* ≤ 0.05 and OR > 1) were puppy dog, expert operator, and successful removal. Variables negatively correlated to excellent time spent for EFBs extraction (*p* ≤ 0.05 and OR < 1) were adult, unspecific clinical signs, and use of polypectomy snare. Variables with *p* ≤ 0.2 and OR > 1 were species, small breed, and older animals. Variables with *p* ≤ 0.2 and OR < 1 were large breed, middle operator, novice operator, use of rat tooth forceps, and multiple forceps. Univariate results related to the excellent time spent for the extraction of EFBs are summarized in Table 8.

Variables negatively correlated to excellent/good time spent for the extraction of EFBs (*p* ≤ 0.05 and OR < 1) were adult animals, localization in stomach (body), and use of polypectomy snare. Variables with *p* ≤ 0.2 and OR > 1 were older animals, localization in stomach (antrum), presence of gastrointestinal clinical signs, and use of laparotomic forceps. Variables with *p* ≤ 0.2 and OR < 1 were large breed, presence of penetrating FBs, localization in cervical esophagus, and use of grasping forceps. Univariate results related to the excellent/good time spent for the extraction of EFBs are summarized in Table 9.

### 3.7. Multivariate Analysis

Variables scoring *p* ≤ 0.20 in a univariate model or considered biologically relevant were included in the multivariable model. Multiple models were considered in relation to the successful removal of EFBs and the time spent for extraction of EFBs (excellent and excellent/good). Models are reported in Table 10.

The use of the combination of polypectomy snare and grasping forceps had, respectively, a 14- and 8-times higher probability of failure to extract the FBs. The successful removal of FBs had a higher probability (22 times) to occur in thirty minutes (excellent time) since the beginning of the endoscopy. In animals with gastrointestinal clinical signs, removal of EFBs had a higher probability (three times) to occur in sixty minutes (excellent/good time) since the beginning of the endoscopy. Variables of adult animals, presence of penetrating FBs, and use of the combination of polypectomy snare and grasping forceps had a lower probability to determine an excellent/good time for the extraction of EFBs.

## 4. Discussion

Our study shows that many factors can influence the successful and timing of the removal of EFBs: medium-breed dogs, adult animals, and FBs localized in the body of the stomach increase the probability of failure in the endoscopic retrieval. Conversely, the success rate and timing of the retrieval of endoscopic FBs are positively influenced in the presence of puppies and experienced operators. Moreover, the use of a combination of devices such as polypectomy snare and grasping forceps negatively influences the successful extraction of EFBs in our population.

In a recent paper, gastrointestinal FBs were most frequently reported in young dogs, male, and medium/large breeds [3]. Esophageal FBs are a common finding in small-breed dogs [2]. In our population, puppies were positively correlated with an excellent time (*p* = 0.002), whereas adult dogs were negatively correlated with an excellent and a good time (*p* = 0.024; *p* = 0.038). Ingestion of FBs is a condition that typically affects young dogs for the exposure to toys and their indiscriminate eating habits [1,8]. The attention of owners for a puppy probably results in early intervention after the ingestion of FBs and consequently in easier and shorter extraction times. Indeed, lots of FBs, plastic objects in particular, can harden on contact with gastric juices and make the passage through the cardia difficult [9]. Moreover, our results show that successful endoscopic removal was negatively correlated to medium breeds (*p* = 0.048). Conversely, large breeds were not associated with an unsuccessful endoscopic removal. Our findings partially disagree with a previous paper [3] and, if this is due to smaller number of medium-breed dogs than large-breed dogs in our study, remains undetermined. We suppose that the size of the animals and increased intragastric space could influence the difficulty of the visualization and extraction of EFBs; however, a large breed was not associated with an unsuccessful endoscopic removal, and this could depend on a greater dilation of the esophageal lumen and the caudal esophageal sphincter.

The most common types of FBs reported in the literature are bone, cartilaginous, and plastic objects [10]. Other FBs frequently ingested by small animals are toys, wooden toothpicks, fishhooks, needles, and food materials [10,11]; however, several FBs are reported in clinical practice [12,13,14,15]. In our study, various types of FBs were detected. In dogs, FBs included mostly socks (12.8%), plastic fragments (12.8%), rags (11.6%), bones (8.1%), pinecones (7%), and wooden toothpicks (5.8%). The detection of needles and fishhooks in our dogs were limited (2.2%). In cats, the FBs that were mainly reported were needles (33.3%), threads (33.3%), shoelaces (16.7%), and stuffed toys (16.7%). There was no statistical difference between the type of FBs, its nature (penetrating or non-penetrating), and success or timing of the removal of EFBs.

Esophageal FBs were detected in 14.1% cases; of these cases, 1.2% were in the cervical esophagus, 10.5% in the thoracic esophagus, and 2.4% in both the thoracic esophagus and stomach. A previous case series revealed that 41% of esophageal FBs had cervical localization [16], and this disagrees with our results. The location of gastric FBs was mostly in the gastric body (32.6%), followed by antrum (18.6%), and both (16.3%) in our dog population. In cats, the location of FBs was mostly in the gastric body (83.3%). In only two animals (2.3%), the localization of FBs was duodenal, and the removal was attempted because FBs were partially located into the stomach. Our results agree with previous studies, in which the upper gastrointestinal tract was mostly affected [3,17]. We speculate that this could be explained by the population of our study, which included only patients submitted to endoscopy and not all patients with gastrointestinal FBs. The authors highlight that the gold standard for the removal of the intestinal FBs is surgical treatment [18]. To the best of authors’ knowledge, no studies have assessed the influence of the localization of FBs during endoscopic retrieval. In addition, our study is the first to divide the gastric site into subcategories, assessing their impact on the endoscopic procedure. Endoscopic time was longer for the gastric body localization (*p* = 0.007); however, there was no statistical difference between the localization of FBs and success of their extraction.

In accordance with other previous studies, dogs and cats with gastrointestinal FBs can be referred with various clinical signs [3,17]. In our study, vomiting was mainly reported (26.1%). In most animals (57.6%) the ingestion of FBs was described by owners, and clinical signs were absent. The attention of owners can lead to early intervention after the ingestion of FBs, preventing the occurrence of serious clinical signs [3]. At the same time, the presence of gastrointestinal clinical signs can alert the owners and encourage them to the veterinary consulting that can result in the rapid diagnosis and treatment of FBs. In our study, a decrease in time spent for endoscopy was associated with presence of gastrointestinal disorders in multivariate models (*p* = 0.033). No specific clinical signs such as depression, ataxia, tremors, tachycardia, and tachypnea were negatively correlated to a short time spent for extraction (*p* = 0.048). These disorders can be associated with dehydration due to gastrointestinal losses and result in the disturbance of fluid balance, acid-based status, and serum electrolyte concentration, with the potential development of anesthetic complications and a time extension of the procedure [19,20].

In the present study, the success rate of the removal of EFBs was 88%. The success rate for the endoscopic retrieval of esophageal FBs ranges from 68 to 88%, whereas the reported rate for gastric foreign bodies ranges from 78 to 94% [4,21]. A questionary published on endoscopic FB retrieval has investigated success rates, outcome, equipment available (endoscope and forceps), type of FBs, and operator’s experience in the different specializations (emergency/general practice/specialist) [4]. There were several limitations to this previous study. Firstly, the outcome of the retrieval of FBs in relation to the operator’s experience was not evaluated. In addition, the operators had different specializations. In our study, we have considered the different levels of experience of specialist endoscopists. The different levels of experience of the clinicians performing the endoscopies influenced the outcome: expert operator was positively correlated to the short time spent for extraction (*p* = 0.005).

To the best of our knowledge, few information is available on the success rates of the various types of ancillary devices for the extraction of EFBs. Ancillary devices were generally used into the working channel of an endoscope. Endoscopic instruments coaxial to the channel of the endoscope as laparotomic forceps or other clips can be used only for esophageal FBs [6]. The use of biopsy forceps and alligator forceps was commonly reported for fishhook retrieval [10]. The use of grasping forceps, Dormia clips, or balloon catheters was reported for esophageal FBs [22,23]. The choice of instruments depended on the endoscopist’s experience, type, and location of FBs. Our results showed the use of various ancillary devices from a minimum of one (60.9%) to maximum of three for the removal of EFBs. The most common type of forceps used were grasping forceps (47.9%), polypectomy snares (41.3%), and alligator forceps (35.8%). Only laparotomic forceps were used coaxially to the operating whip in 13.1% of all cases and, specifically, in 7/12 (58.3%) dogs with esophageal FBs. These forceps are useful only for esophageal FBs because of their length and flexibility, unable to reach the stomach. However, the laparotomic forceps has a higher tensile force due to its larger size, so it allows better anchoring on FBs [6,23]. The use of a polypectomy snare in addition to grasping forceps causes an unsuccessful FB removal and an increase in time for endoscopy in our study. Conversely, some authors recommended using a polypectomy snare for the removal of esophageal or gastric EFBs [24]. We speculate that this could be influenced by our staff preference in using these forceps more frequently. However, the characteristics of the FBs can also influence the success of the clamps used. Grasping forceps is formed by small arms that have a hook at the end. These characteristics make it useful in the presence of FBs with irregular surfaces such as pinecones or textile objects. Moreover, the surface of FBs can become slippery because of gastric mucous film resulting in less grip surface for the clamp. Additionally, plastic FBs become harder during chronic persistence in the stomach and more resistant to the penetration of the branches of forceps [9]. Also, textiles can fray because of the imbibition of FBs by gastric juices or in the presence of a fine texture. These factors could determine the failure of the grasping forceps despite the appearance of FBs being favorable to its use. Moreover, the use of multiple devices was negatively correlated with an excellent time for endoscopy removal (*p* = 0.002) in our study. This could be related to greater difficulty in extraction and, therefore, the use of multiple devices increased the time needed for the procedure.

In the present study, the mean time spent for endoscopic extraction was 54.79 min (range, 10–120 min). Based on responses from veterinarians, Wood et al. (2021) defined a maximum time limit of 60 min for the retrieval of gastric EFBs and no time limit for the removal of esophageal EFBs, although the attempted removal should last at most 60 min [4]. Absence of a maximum time limit for the retrieval of esophageal EFBs depends on the need for surgical removal in the event of failure, except in cases where digestible material can be pushed into the stomach. The surgical removal of esophageal FBs has longer hospitalization and an increased risk of death in dogs [21]. The time spent on endoscopy depends on the difficultly of the extraction of the FBs, operator’s experience, and clinical conditions of the patients. A reduced time for endoscopy is useful for decreasing the time of anesthesia and improve the outcome in esophageal obstructions by FBs. Indeed, esophageal obstructions should be considered an emergency because of the higher the risk of necrosis, ulceration, and perforation of the esophageal mucosa, which can lead to pneumothorax, pneumomediastinum, or pyothorax if FBs remain for a long time [21].

In the present study, complications were reported in 43.3% of animals and were classified as minor (92.5%) or major (7.5%). Complications can be associated with the irregular surface of FBs (penetrating FBs), to compression of FBs, or to entrapment of esophageal FBs [2,22]. In our cases, major complications were associated with esophageal FBs and were observed only in small dogs with penetrating FBs (bone) in the thoracic esophagus. In previous studies, poor prognosis and severe complications were reported in small dogs, when the esophageal obstruction was present for a long time, and in the presence of FBs as a bone and fishhook [2,21,25]. This agrees with our results. Only one dog (1.1%) died in our study because of esophageal perforation and pneumothorax during endoscopic foreign body extraction. A previous study reports a mortality rate of 5.4% for esophageal FBs [2].

The limitations of our study are related to its retrospective nature. A low number of cats were present in the study. The different levels of experience of the operators performing the endoscopy influenced the outcome, although this represents a “real world” scenario. Finally, no information has been collected on time of hospitalization or the development of complications like esophageal stenosis due to the retrieval of esophageal EFBs [2,26].

## 5. Conclusions

Endoscopy is a minimally invasive technique with a high success rate, and it represents the first choice for removing FBs in the upper gastrointestinal tract. Our study shows that the success rate and time for the removal of EFBs can be influenced by several factors. Additional prospective and comparative studies in a large and multicentric population of patients undergoing the removal of EFBs are needed to further evaluate potential factors that influence its success and to create interventional endoscopic guidelines, as in human medicine [27].

## Figures and Tables

**Figure 1 vetsci-10-00560-f001:**
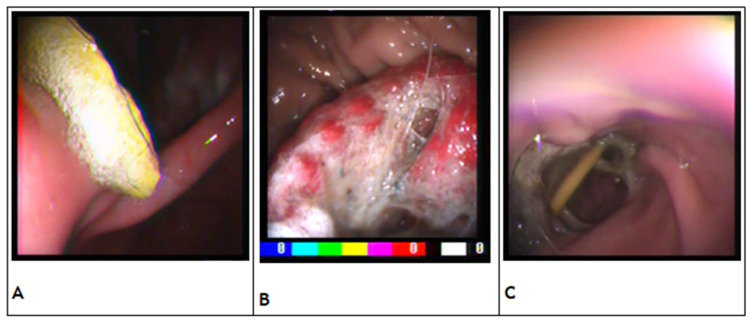
(**A**) Plastic FB in the fundus of the stomach; (**B**) textile FB in the body of the stomach; (**C**) wooden toothpick in the antrum of the stomach.

**Figure 2 vetsci-10-00560-f002:**
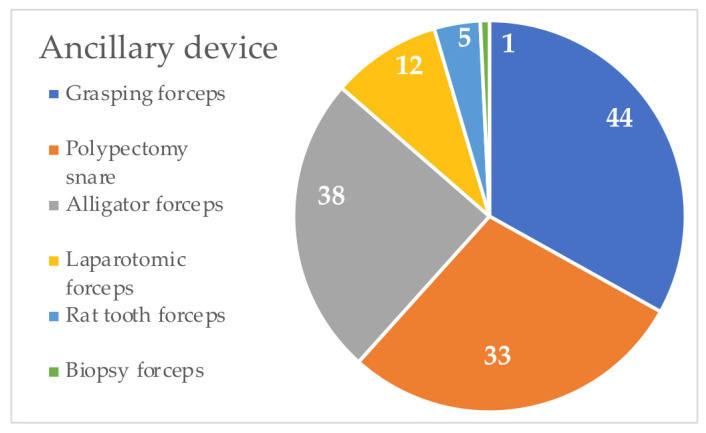
Graph shows the number of cases in which ancillary devices were used for endoscopic removal of esophageal and gastric FBs.

**Figure 3 vetsci-10-00560-f003:**
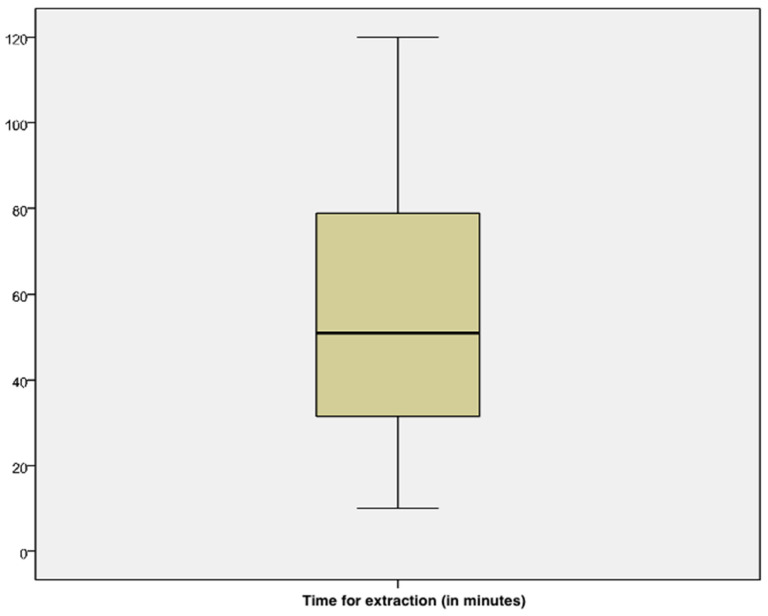
Boxplot shows time spent for the extraction of FBs in the animals included in the study.

**Figure 4 vetsci-10-00560-f004:**
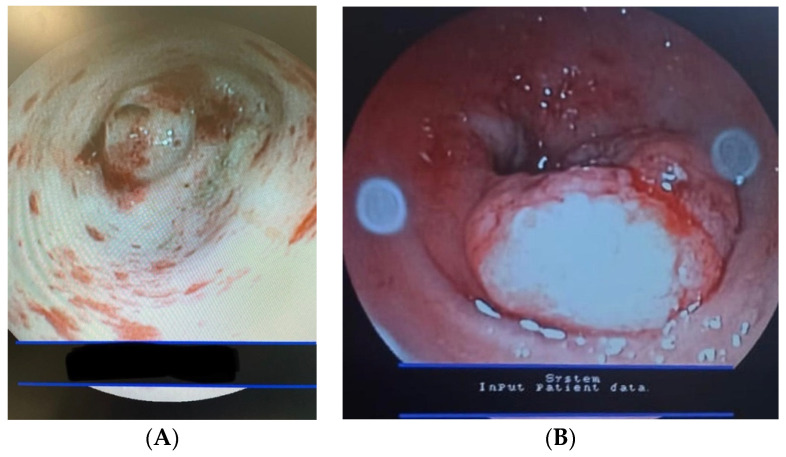
Main complications observed in dogs with esophageal FBs. (**A**). Thoracic esophagus—esophageal mucosa was characterized by 2–4 mm multifocal erosions/ulcerations and diffuse hemorrhages. (**B**)**.** Cardia (thoracic esophagus)—esophageal mucosa shows edema, hyperemia, and hemorrhages. A 4 mm mucosal erosion/ulceration is evident.

**Table 1 vetsci-10-00560-t001:** Table shows frequency and percent of breeds of the dogs included in the study.

Breeds	Frequency	Percent
Mixed-breed dog	23/86	26.7%
German Shepherd	6/86	7%
Bernese Mountain Dog	5/86	5.8%
Dachshund, Cocker Spaniel, Jack Russel Terrier	4/86	4.7%
Beagle, Boxer, Golden Retriever, Labrador Retriever, Hound	3/86	3.5%
Great Dane, French Bulldog, Bullmastiff, Chihuahua, English Setter, Spitz	2/86	2.3%
Poodle, Akita Inu, Border Collie, Corso, Dobermann, Fox Terrier, Husky, Maremma Sheepdog, Belgian Shepherd, Pekingese, Schnauzer, Springer Spaniel, West Highland White Terrier	1/86	1.2%

**Table 2 vetsci-10-00560-t002:** Table shows clinicals signs described by owners or observed at presentation.

Gastrointestinal Disorders	Frequency	Percent
Vomiting	24/92	26.1%
Anorexia	3/92	3.3%
Abdominal pain	3/92	3.3%
Inappetence	2/92	2.2%
Dysphagia	2/92	2.2%
Pica	1/92	1.1%
Regurgitation	1/92	1.1%
**Unspecific Clinical Signs**	**Frequency**	**Percent**
Depression	4/92	4.4%
Congestion of the mucosae membranes	2/92	2.2%
Weight loss	2/92	2.2%
Ataxia	1/92	1.1%
Tremors	1/92	1.1%
Polyuria and polydipsia	1/92	1.1%
Tachycardia	1/92	1.1%
Tachypnea	1/92	1.1%
**Respiratory Disorder**	**Frequency**	**Percent**
Dyspnea	2/92	2.2%
Cough	1/92	1.1%
Stridor	1/92	1.1%

**Table 3 vetsci-10-00560-t003:** Table shows the localization of esophageal and gastric foreign bodies in dogs included in the study.

Localization	Frequency	Percent
Cervical esophagus	1/86	1.2%
Thoracic esophagus	9/86	10.5%
Stomach (fondus)	10/86	11.6%
Stomach (body)	28/86	32.6%
Stomach (antrum)	16/86	18.6%
Thoracic esophagus + Stomach (body)	1/86	1.2%
Stomach (fondus) + Stomach (body)	14/86	16.3%
Stomach (fondus) + Stomach (antrum)	1/86	1.2%
Stomach (body) + Stomach (antrum)	2/86	2.3%
Stomach (antrum) + Duodenum	2/86	2.3%
Thoracic esophagus + Stomach (fundus) + Stomach (body)	1/86	1.2%
Stomach (fundus) + Stomach (body) + Stomach (antrum)	1/86	1.2%

**Table 4 vetsci-10-00560-t004:** Table shows clinical and endoscopic findings in cats with FBs. Similar findings were underlined.

	Signalment	Clinical Signs	FBs (Number, Type, and Localization)	Endoscopic Removal
1.	Domestic Shorthair cat, female neutered, 24 months	Ingestion of FBs was seen by owners	One needle (penetrating FB) in the stomach (body)	Operator’s experience: expert Single endoscopic instrument used (Alligator forceps) Extraction time: 79 min. Successful removal Complications: minor (gastritis)
2.	Domestic Shorthair cat, female neutered, 6 months	Gastrointestinal disorders: abdominal pain	One shoelace (non-penetrating FB) in the stomach (body)	Operator’s experience: expert Single endoscopic instrument used (Grasping forceps) Extraction time: 19 min. Successful removal Complications: minor (gastritis)
3.	Domestic Shorthair cat, female, 12 months	Gastrointestinal disorders: vomiting	One thread (non-penetrating) in the stomach (body)	Operator’s experience: novice Single endoscopic instrument used (Grasping forceps) Extraction time: 26 min. Successful removal Complications: minor (gastritis)
4.	Bengal cat, female, 12 months	Ingestion of FBs was seen by owners	One needle (penetrating) in the stomach (body)	Operator’s experience: middle Single endoscopic instrument used (Alligator forceps) Extraction time: 54 min. Successful removal Complications: minor (gastritis)
5.	Domestic Shorthair cat, female, 24 months	Ingestion of FBs was seen by owners	One stuffed toy (non-penetrating) in the stomach (antrum)	Operator’s experience: middle Single endoscopic instrument used (Polypectomy snare) Extraction time: 52 min. Successful removal Complications: minor (gastritis)
6.	Domestic Shorthair cat, female, 48 months	Ingestion of FBs was seen by owners	One thread (non-penetrating) in the stomach (body)	Operator’s experience: middle Single endoscopic instrument used (Alligator forceps) Extraction time: 15 min. Successful removal Complications: minor (gastritis)

**Table 5 vetsci-10-00560-t005:** Ancillary devices used for endoscopic removal of gastrointestinal FBs.

Ancillary Devices	Frequency
Grasping forceps	44/92
Polypectomy snare	33/92
Alligator forceps	38/92
Laparotomic forceps	12/92
Rat tooth forceps	5/92
Biopsy forceps	1/92

**Table 6 vetsci-10-00560-t006:** Table shows clinical and endoscopic findings in dogs with esophageal FBs. Minor complications were underlined, whereas major complications were in red. Similar findings for major complications were in bold.

	Signalment	Clinical Signs	FBs (Type, Number, and Localization)	Endoscopic Removal
1.	Breed: mixed breed Size: small breed Sex: male Age: 180 months	Ingestion of FBs was seen by owners	Type: aluminum foil for aliments, synthetic casing for aliments FBs number: 2 Localization: thoracic esophagus	Operator’s experience: expert Endoscopic instruments used: rat tooth forceps and laparotomic forceps Extraction time: 88 min. Successful removal Complications: no
2.	Breed: Spitz Size: small breed Sex: female Age: 60 months	Ingestion of FBs was seen by owners	Type: cookie FBs number: 1 Localization: thoracic esophagus	Operator’s experience: expert Endoscopic instruments used: alligator forceps and laparotomic forceps Extraction time: 53 min. Successful removal Complications: no
3.	Breed: Bullmastiff Size: large breed Sex: male Age: 60 months	Unspecific clinical signs: weight loss Gastrointestinal signs: vomiting	Type: rag FBs number: 1 Localization: thoracic esophagus	Operator’s experience: expert Endoscopic instruments used: alligator forceps and laparotomic forceps Extraction time: 48 min. Successful removal Complications: minor (esophagitis)
4.	Breed: West Highland Withe Terrier Size: small breed Sex: male Age: 168 months	Respiratory disorders: dyspnea	Type: apple core FBs number: 1 Localization: thoracic esophagus	Operator’s experience: middle Endoscopic instruments used: alligator forceps and laparotomic forceps Time extraction: 45 min. Successful removal Complications: no
5.	Species: dog Breed: mixed breed Size: small breed Sex: male Age: 120 months	Ingestion of FBs was seen by owners	Type: cartilaginous tissue FBs number: 1 Localization: thoracic esophagus	Operator’s experience: expert Endoscopic instruments used: laparotomic forceps Time extraction: 30 min. Successful removal Complications: no
6.	Species: dog Breed: mixed breed **Size: small breed** Sex: male Age: 24 months	Gastrointestinal disorders: inappetence, regurgitation	**Type: bone** FBs number: 1 **Localization: thoracic esophagus**	Operator’s experience: middle Endoscopic instruments used: laparotomic forceps Time extraction: 56 min. Successful removal **Complications: major (perforation, pneumothorax, death**)
7.	Species: dog Breed: Dachshund Size: small breed Sex: female Age: 204 months	Ingestion of FBs was seen by owners	Type: bone FBs number: 1 Localization: thoracic esophagus	Operator’s experience: middle Endoscopic instruments used: alligator forceps Time extraction: 29 min. Successful removal Complications: no
8.	Species: dog Breed: mixed breed **Size: small breed** Sex: male Age: 192 months	Unspecific clinical signs: depression, polyuria, polydipsia Gastrointestinal disorders: anorexia, vomiting	**Type: bone** FBs number: 1 **Localization: thoracic esophagus**	Operator’s experience: expert Endoscopic instruments used: grasping forceps, laparotomic forceps Time extraction: 50 min. Successful removal **Complications: major (erosion/ulceration, needs to PEG)**
9.	Species: Dog Breed: Mixed breed Size: Small breed Sex: Female Age: 180 months	Respiratory disorders: Cough, dyspnea	Type: Bone FBs number: 1 Localization: Thoracic esophagus	Operator’s experience: Middle Endoscopic instruments used: Laparotomic forceps Time extraction: 19 min. Successful removal Complications: Minor (esophagitis)
10	Species: dog Breed: Jack Russel Terrier **Size: small breed** Sex: male Age: 60 months	Gastrointestinal disorders: dysphagia	**Type: bone** FBs number: 1 **Localization: thoracic esophagus**	Operator’s experience: novice Endoscopic instruments used: polypectomy snare Time extraction: 86 min. Successful removal **Complications: major (erosion/ulceration)**
11	Species: dog Breed: Labrador Retriever Size: large breed Sex: male Age: 4 months	Ingestion of FBs was seen by owners	Type: fishhook FBs number: 1 Localization: cervical esophagus	Operator’s experience: middle Endoscopic instruments used: alligator forceps Time extraction: 79 min. Successful removal Complications: minor (esophagitis)
12.	Species: dog Breed: mixed breed Size: small breed Sex: male Age: 6 months	Gastrointestinal disorders: dysphagia, vomiting	Type: wood FBs number: 1 Localization: thoracic esophagus	Operator’s experience: middle Endoscopic instruments used: polypectomy snare and grasping forceps Time extraction: 89 min. Successful removal Complications: minor (esophagitis)

**Table 7 vetsci-10-00560-t007:** Results of univariate analysis related to the success rate of the removal of EFBs. Variables with *p* ≤ 0.05 and OR > 1 were significatively and positively associated with the removal of EFBs. Variables with *p* ≤ 0.05 and OR < 1 were significatively and negatively associated with the removal of EFBs. Variables with *p* ≤ 0.2 were used for multivariate analysis. OR = odds ratios; IC = confidence interval.

Variable	*p* Value ≤ 0.05	OR > 1	95% IC
Good time	0.035	7.234	0.884–59.220
**Variable**	***p* Value ≤ 0.05**	**OR < 1**	**95% IC**
Medium breed	0.048	0.843	0.762–0.933
Polypectomy snare	0.001	0.094	0.019–0.466
Grasping forceps	0.016	0.169	0.034–0.832
Multiple forceps	0.002	0.111	0.022–0.551
Excellent time	0.048	0.843	0.762–0.933
Not acceptable time	0.000	0.067	0.012–0.363
**Variable**	***p* Value ≤ 0.2**	**OR > 1**	**95% IC**
Expert operator	0.148	3.094	0.628–15.247
**Variable**	***p* Value ≤ 0.2**	**OR < 1**	**95% IC**
Large breed	0.169	0.417	0.117–1.489
Adult	0.151	0.393	0.106–1.450
Thoracic esophagus	0.193	0.864	0.793–0.942
Middle operator	0.175	0.413	0.112–1.525
Laparotomic forceps	0.171	0.863	0.790–0.941
Acceptable time	0.064	0.311	0.086–1.122

**Table 8 vetsci-10-00560-t008:** Results of univariate analysis related to the excellent time spent for extraction of EFBs. Variables with *p* ≤ 0.05 and OR > 1 were significatively and positively associated with time of the removal of EFBs. Variables with *p* ≤ 0.05 and OR < 1 were significatively and negatively associated with time of the removal of EFBs. Variables with *p* ≤ 0.2 were used for multivariate analysis. OR = odds ratios; IC = confidence interval.

Variable	*p* Value ≤ 0.05	OR > 1	95% IC
Puppy	0.002	8.375	1.889–37.136
Expert operator	0.005	4.083	1.490–11.188
Successful removal	0.048	1.373	1.202–1.568
**Variable**	***p* Value ≤ 0.05**	**OR < 1**	**95% IC**
Adult	0.024	0.294	0.098–0.885
Unspecific clinical signs	0.048	0.728	0.638–0.832
Polypectomy snare	0.047	0.314	0.096–1.026
**Variable**	***p* Value ≤ 0.2**	**OR > 1**	**95% IC**
Species	0.121	3.526	0.658–18.910
Small breed	0.116	2.160	0.818–5.705
Older	0.161	2.250	0.711–7.124
**Variable**	***p* Value ≤ 0.2**	**OR < 1**	**95% IC**
Large breed	0.141	0.441	0.146–1.333
Middle operator	0.168	0.494	0.180–1.360
Novice operator	0.068	0.175	0.022–1.406
Rat tooth forceps	0.197	0.747	0.661–0.844
Multiple forceps	0.071	0.370	0.123–1.115

**Table 9 vetsci-10-00560-t009:** Results of univariate analysis related to the excellent/good time spent for extraction of EFBs. Variables with *p* ≤ 0.05 and OR > 1 were significatively and positively associated with removal of EFBs. Variables with *p* ≤ 0.05 and OR < 1 were significatively and negatively associated with removal of EFBs. Variables with *p* ≤ 0.2 were used for multivariate analysis. OR = odds ratios; IC = confidence interval; FB = foreign body; GI = gastrointestinal.

Variable	*p* Value ≤ 0.05	OR < 1	95% IC
Adult	0.038	0.405	0.171–0.961
Stomach (body)	0.007	0.290	0.116–0.728
Polypectomy snare	0.047	0.413	0.172–0.995
**Variable**	***p* Value ≤ 0.2**	**OR > 1**	**95% IC**
Older	0.080	3.152	0.829–11.981
Stomach (antrum)	0.063	2.769	0.923–8.313
GI clinical signs	0.112	2.101	0.835–5.285
Laparotomic forceps	0.102	3.511	0.722–17.080
**Variable**	***p* Value ≤ 0.2**	**OR < 1**	**95% IC**
Large breed	0.123	0.505	0.211–1.210
Penetrating FBs	0.053	0.430	0.181–1.020
Cervical esophagus	0.199	0.374	0.286–0.487
Grasping forceps	0.067	0.451	0.191–1.064

**Table 10 vetsci-10-00560-t010:** Multivariate models showing different relationships among variables selected from the univariate models. All variables present in these models were statistically relevant (*p* ≤ 0.05). The OR and respective CI 95% represent the probability of outcome (failure to remove EFBs and time spent for extraction of EFBs, excellent and excellent/good) of individual variables in the groups when they were present at the same time. OR = odds ratios; IC = confidence interval; FB = foreign body; GI = gastrointestinal.

Outcome = Failure to Remove			
Variable	*p* Value	OR	95% IC
Polypectomy snare	0.002	14.286	2.644–77.176
Grasping forceps	0.014	8.514	1.545–46.917
Outcome = time excellent for extraction			
Successful removal	0.004	22.400	2.718–184.574
Outcome = time excellent/good for extraction			
Adult	0.003	0.179	0.058–0.551
Penetrating FBs	0.010	0.241	0.081–0.713
GI clinical signs	0.033	3.517	1.107–11.171
Polypectomy snare	0.032	0.314	0.109–0.906
Grasping forceps	0.003	0.173	0.056–0.543

## Data Availability

The data presented in this study are available in the article.

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
