# Peer review of "Endoscopic Retrieval of Esophageal and Gastric Foreign Bodies in Cats and Dogs: A Retrospective Study of 92 Cases"

_vetsci, 2023, doi:10.3390/vetsci10090560_

Round 1

Reviewer 1 Report

General comments:

Although this is a very interesting and important topic in the veterinary clinical setting, this manuscript has significant revisions necessary to be considered for publication.

It is a retrospective study that has both cats and dogs, so for that reason, it has to be better documented. I suggest to add figures, images and more detailed data of the clinical cases.

There are few bibliographic references and this is a very discussed subject. Authors need to increase their bibliographic research, in the introduction and mostly in the discussion section. This particular section has to be improved.

There is a clear absence of study design that should be better explained to the read and clearer in the manuscript.

The authors should consider these major revions and submit for publication after these alterations. 

Author Response

We thank the Reviewer for her/his insightful comments and suggestions. We have addressed all the comments raised and we hope that our changes and responses below will satisfy the Reviewer.

Reviewer 1:

General comments:

Although this is a very interesting and important topic in the veterinary clinical setting, this manuscript has significant revisions necessary to be considered for publication.

We thank the Reviewer for her/his encouraging comments.

It is a retrospective study that has both cats and dogs, so for that reason, it has to be better documented. I suggest to add figures, images and more detailed data of the clinical cases.

We thank the Reviewer for her/his suggestions. As the Reviewer can see, we have added 2 new tables on clinical and endoscopic findings and the Figure 2. We have also added more data following the suggestions of the Reviewer 2.

There are few bibliographic references and this is a very discussed subject. Authors need to increase their bibliographic research, in the introduction and mostly in the discussion section. This particular section has to be improved.

We thank the Reviewer for her/his suggestions. We have increased the references and we have improved the Introduction and Discussion.

There is a clear absence of study design that should be better explained to the read and clearer in the manuscript.

Our study is a retrospective study and its aim is to reported our experience in the endoscopic retrieval of esophageal and gastric foreign bodies in dogs and cats. We have evaluated several factors that can influence the success rate and timing for FBs endoscopic retrieval. We have also undelighted the correlation between the operator’s experience and the success rate for the EFBs removal.

We believe that data reported can be useful for increasing information regarding this subject and aiding in the formation of future guidelines.

We have better reported this in the manuscript following the suggestions of both Reviewers.

The authors should consider these major revisions and submit for publication after these alterations.

We thank again the Reviewer for her/his suggestions.

Reviewer 2 Report

The study is well writen with a nice number of cases, however none big news for vet litterature. 

I have few comments an d suggestions.

Notes:

-Editing of English language required

Abstract :

-esophageal (12) and gastric (90) FB. Total 102? I suppose there are some combined esophageal and gastric. 

- Time od surgery 59,74 minutes . Please add range of time.

- Endoscopic FB should be EFBs. Please use acronyms in all text.

Introduction:

- Please use acronymus in all text

M&M:

- Esophageal (12) and gastric (90). Total 102?. I think there are combined cases esophageal-gatsric.

- Any preoperative Radiography or CT scan exams?

- Please age in months also for adult and older animals.

- I don't understand traumatic vs atraumatic. You mean penetrating vs non-penetrating?

- You didn't mentioned the muscosa status after FB removal. I think this is very important for the outcome/complication and survival of the animals.

- Why Duodenal ? The inclusion criteria is gastric. And after you described that the last location of the FB is Antrum of the stomach.

- The experiences of endoscopy should be evaluated in the number of  cases not years . This is the main limitation of the study. Furthermore , there are any Board-certificated (European or American Surgeon or Internal medicine) that did the procedures? I think that you must described the experiences in number of procedures not year, or if the operator are board-certificated or not. 

- Complication rate (minor or major)?

Results:

- Please use Domestic Shorthair cat , not European cat. 

- In 92 cases there are not any neutered female or neutered male? Please recheck.

- Please, age in months

- Pits ? 

- Any fishhook or needle? 

- Again, I don't understand traumatic or atraumatic. I think should be very important to know how many cases have penetrating or not penetrating injury. 

- How many cases convert in surgical procedures?

- Table 2. Here you described combined esophageal and gastric FBs. Should be reported in abstract and MeM. (see comment before)

Cats:

- Again, I think that penetrating vs penetrating injury should be described. Also the dimension of the lesion and rate of conversion.

Clinical signs:

Please move the clinical signs paragraph before. Should be the first paragraph of result's description.

Edoscopic removal:

- Please write:  88% (81/92) of animals [75/86 dogs (87,2%) 6/6 cats (100%)]

- Please see my comment before about "experiences of surgeon"

- Any information if the instruments are used in the working channel of endoscope or coaxially. This could be interesting.

Discussion:

- Any information about use the instrument coaxially or in the channel of endoscope?

Add Citation:

Brisson A. et al . Risk factor and prognostic indicators for surgical outcome of dogs with esophageal foreign body obstructions. JAVMA 2018 301-308

Sterman A. Likelihood and outcome of esophageal perforation secondary to esophageal foreign body in dogs. JAVMA 2018, 1053-1056

Moderate Editing of English is required

Author Response

We thank the Reviewer for her/his insightful comments and suggestions. We have addressed all the comments raised and we hope that our changes and responses below will satisfy the Reviewer.

Reviewer 2:

The study is well written with a nice number of cases, however none big news for vet literature.

I have few comments and suggestions.

We thank the Reviewer for her/his encouraging comments. In this retrospective study, we have reported our experience in the endoscopic retrieval of esophageal and gastric foreign bodies in dogs and cats. We have evaluated several factors that can influence the success rate and timing for FBs endoscopic retrieval. We have also undelighted the correlation between the operator’s experience and the success rate for the EFBs removal. We believe that data reported can be useful for increasing information regarding this subject and aiding in the formation of future guidelines.

Notes:

-Editing of English language required

We have carefully edited the English language.

Abstract:

-esophageal (12) and gastric (90) FB. Total 102? I suppose there are some combined esophageal and gastric.

We thank the Reviewer for her/his suggestion. There is a typo in the manuscript. The number of gastric FBs is 82 and as the Reviewer can see in table 3, 2 dogs have FBs in both the esophagus and stomach.

We have amended the sentences, as follows: “Medical records of 92 animals undergoing endoscopic removal of esophageal (n=12) and gastric (n=84) FBs have been reviewed. Two dogs had FBs in both the esophagus and stomach.” We have better explained what the Reviewer suggested in the Abstract and Materials & Methods.

- Time on surgery 59,74 minutes. Please add range of time.

Done

- Endoscopic FB should be EFBs. Please use acronyms in all text.

Done

Introduction:

- Please use acronymus in all text

Done

M&M:

- Esophageal (12) and gastric (90). Total 102?. I think there are combined cases esophageal-gastric.

As reported above, we have specified, as follows: ”Medical records of 92 animals undergoing endoscopic removal of esophageal (n=12) and gastric (n=84) FBs have been reviewed. Two dogs had FBs in both the esophagus and stomach.”

- Any preoperative Radiography or CT scan exams?

We have added the following sentence in the Results: “In all cases radiographic and/or ultrasonographic examinations were performed by referring veterinarian or in the Hospital/Clinic of the authors, and their results suggested the presence of esophageal or gastrointestinal FBs.“

- Please age in months also for adult and older animals.

Done

- I don’t understand traumatic vs atraumatic. You mean penetrating vs non-penetrating?

We have changed the sentence following the Reviewer’s suggestion: “Based on their potential for damaging the gastrointestinal mucosa, FBs were classified in non-penetrating when characterized by smooth outer surface, and in penetrating when characterized by irregular or sharp surface.”

- You didn’t mentioned the muscosa status after FB removal. I think this is very important for the outcome/complication and survival of the animals.

We have added the information requested from the Reviewer: “Moreover, data on type, number, and location of FBs, endoscopic instruments, time spent to the extraction or spent attempting FB retrieval before recommending surgery, and mucosa status after EFBs removal were collected by reviewing medical records and video recordings.”

- Why Duodenal ? The inclusion criteria is gastric. And after you described that the last location of the FB is Antrum of the stomach.

We thank the Reviewer for this suggestion. We have removed duodenal and we have better explained in the manuscript as follows: ”In 2 dogs (2,3%) FBs were localized in stomach extending into the duodenum.”

- The experiences of endoscopy should be evaluated in the number of  cases not years . This is the main limitation of the study. Furthermore , there are any Board-certificated (European or American Surgeon or Internal medicine) that did the procedures? I think that you must described the experiences in number of procedures not year, or if the operator are board-certificated or not.

We have followed the Reviewer’s suggestion and we have changed the sentence as follows: “Based on experience, operators were classified in novice (<50 endoscopic examinations for year), middle (between 50 and 150 endoscopic examinations for year) and expert (> 150 endoscopic examinations for year).”

- Complication rate (minor or major)?

We have added the information requested from the Reviewer, as follows: ”Complications were categorized into minor and major. Minor complications were characterized by inflammation of esophageal or gastrointestinal mucosa with evident hyperemia, edema, and minor bleeding. Major complications were characterized by mucosal erosion/ulceration, application of Percutaneous Endoscopic Gastrostomy (PEG), wall perforation and death.”

Results:

- Please use Domestic Shorthair cat , not European cat.

Done

- In 92 cases there are not any neutered female or neutered male? Please recheck.

We have added the information requested, as follows: “Twenty-eight dogs were females (32,6%) and 57 were males (66,3%). Dogs neutered were 11/86 (30,2%), 6 (21,4%) females and 5 (8,8%) males. All cats were females and 33,3% of these were neutered (2/6).”

- Please, age in months

Done

- Pits ?

We have changed the word with “pinecones”.

- Any fishhook or needle?

We have added the information requested, as follows: “EFBs were socks (11/86 [12,8%]), plastics fragments (11/86 [12,8%]), rags (10/86 [11,6%]), bones (7/86 [8,1%]), pinecones (6/86 [7%]), wooden toothpick (5/86 [5,8%]), needle (2/86 [2,2%]), fishhook (2/86 [2,2%]), and other FBs (range 1 - 4 [1,1 - 4,3 %]).”

- Again, I don't understand traumatic or atraumatic. I think should be very important to know how many cases have penetrating or not penetrating injury.

As reported above, we have changed the classification of the FBs.

- How many cases convert in surgical procedures?

We have specified in the manuscript: “In 11/92 (12%) patients, surgical intervention was required to safely remove FBs.”.

- Table 2. Here you described combined esophageal and gastric FBs. Should be reported in abstract and MeM. (see comment before)

Done

Cats:

- Again, I think that penetrating vs penetrating injury should be described. Also the dimension of the lesion and rate of conversion.

We have used the classification suggested from the Reviewer. We have also added the Table 5 with clinical and endoscopic findings in cats.

Clinical signs:

Please move the clinical signs paragraph before. Should be the first paragraph of result's description.

We have moved the paragraph above in the manuscript but after the paragraph “Animals”. We believe that “Clinical signs” should be after the paragraph “Animals”.

Endoscopic removal:

- Please write:  88% (81/92) of animals [75/86 dogs (87,2%) 6/6 cats (100%)]

Done

- Please see my comment before about “experiences of surgeon”

We have changed the criteria for the operator’s experience.

- Any information if the instruments are used in the working channel of endoscope or coaxially. This could be interesting.

We have changed the sentence as follows: “Ancillary devices (polypectomy snare, rat tooth forceps, alligator forceps, grasping forceps, biopsy forceps) were used within the working channel of the endoscope, whereas laparotomic forceps were used coaxially to the operating whip.”

Discussion:

- Any information about use the instrument coaxially or in the channel of endoscope?

We have added some information following the Reviewer suggestion: “Ancillary devices were used generally into the working channel of endoscope. Endoscopic instruments coaxially to channel of endoscope as laparotomic forceps or other clips can be used only for esophageal FBs [6]. Use of biopsy forceps and alligator forceps was commonly reported for fishhook retrieval [10]. Use of grasping forceps, Dormia clips, or balloon catheter was reported for esophageal FBs [22,23]. The choice of instruments depended on the endoscopist’s experience, type and location of FBs. Our results showed the use of various ancillary devices from minimum of one (60,9%) to maximum of three for FBs removal. The most common type of forceps used were grasping forceps (47,9%), polypectomy snare (41,3%), and alligator forceps (35,8%). Only laparotomic forceps were used coaxially to the operating whip in 13,1% of all cases and specifically, in 7/12 (58,3%) dogs with esophageal FBs. This forceps is useful only for esophageal FBs because of its length and flexibility, unable to reach the stomach. However, the laparotomic forceps has a higher tensile force due to its larger size, so it allows better anchoring on foreign bodies [6,23].”

Add Citation:

Brisson A. et al . Risk factor and prognostic indicators for surgical outcome of dogs with esophageal foreign body obstructions. JAVMA 2018 301-308

Sterman A. Likelihood and outcome of esophageal perforation secondary to esophageal foreign body in dogs. JAVMA 2018, 1053-1056

Done

Round 2

Reviewer 1 Report

The authors cope with all the suggestions made, and I think the manuscript was improved, being acceptable for publication.